# Genome-Wide Identification of *CHYR* Gene Family in *Sophora alopecuroides* and Functional Analysis of *SaCHYR4* in Response to Abiotic Stress

**DOI:** 10.3390/ijms25116173

**Published:** 2024-06-04

**Authors:** Youcheng Zhu, Ying Wang, Zhipeng Ma, Di Wang, Fan Yan, Yajing Liu, Jingwen Li, Xuguang Yang, Ziwei Gao, Xu Liu, Le Wang, Qingyu Wang

**Affiliations:** 1College of Biological and Agricultural Engineering, Jilin University, Changchun 130022, China; youchengzhu@jlu.edu.cn; 2College of Plant Science, Jilin University, Changchun 130062, China; wangying2009@jlu.edu.cn (Y.W.); mazp20@mails.jlu.edu.cn (Z.M.); wdi22@mails.jlu.edu.cn (D.W.); yanfan@jlu.edu.cn (F.Y.); yj_liu@jlu.edu.cn (Y.L.); jingwen@jlu.edu.cn (J.L.); xgyang@jlu.edu.cn (X.Y.); zwgao22@mails.jlu.edu.cn (Z.G.); xul23@mails.jlu.edu.cn (X.L.)

**Keywords:** *Sophora alopecuroides*, *CHYR* gene, abiotic stress, genome-wide identification, expression profile

## Abstract

*Sophora alopecuroides* has important uses in medicine, wind breaking, and sand fixation. The CHY-zinc-finger and RING-finger (CHYR) proteins are crucial for plant growth, development, and environmental adaptation; however, genetic data regarding the *CHYR* family remain scarce. We aimed to investigate the *CHYR* gene family in *S. alopecuroides* and its response to abiotic stress, and identified 18 new *SaCHYR* genes from *S. alopecuroides* whole-genome data, categorized into 3 subclasses through a phylogenetic analysis. Gene structure, protein domains, and conserved motifs analyses revealed an exon–intron structure and conserved domain similarities. A chromosome localization analysis showed distribution across 12 chromosomes. A promoter analysis revealed abiotic stress-, light-, and hormone-responsive elements. An RNA-sequencing expression pattern analysis revealed positive responses of *SaCHYR* genes to salt, alkali, and drought stress. *SaCHYR4* overexpression considerably enhanced alkali and drought tolerance in *Arabidopsis thaliana*. These findings shed light on SaCHYR’s function and the resistance mechanisms of *S. alopecuroides*, presenting new genetic resources for crop resistance breeding.

## 1. Introduction

*Sophora alopecuroides*, a perennial shrub belonging to the Leguminosae family, is known for its well-developed roots and remarkable resilience in arid and sandy soils. This particular species exhibits a wide range of practical applications in medicine, pesticides, fodder, nectar production, windbreaks, and sand fixation [1]. In the field of medicine, *S. alopecuroides* has heat elimination and detoxification functions, as well as antibacterial, anti-inflammatory, analgesic, and insecticidal properties [1,2].

Various compounds isolated from *S. alopecuroides*, including alkaloids, flavonoids, and steroids, play important roles in the treatment of gynecological inflammation and exhibit neuroprotective and anticancer properties [2,3]. In particular, the alkaloids in *S. alopecuroides* have been used in pest control against, for example, diamond moths, cinnabar mites, red spiders, and aphids and to regulate plant growth activities, such as promoting the growth and formation of cucumber cotyledons and roots [2,4]. Alkaloid extraction yields a high-protein residue that can be developed into a high-protein feed [1,4].

Known as the “desert honey reservoir”, *S. alopecuroides* is also an important nectar source, owing to its rich flowers, delicate nectar, long flowering period, and high nutritional value [1]. Of note, *Sophora alopecuroides* is resistant to salt, alkali, drought, and wind erosion and can improve soil and maintain the ecological balance in arid and semi-arid areas, which is vital for preventing and controlling desertification [1,5,6]. This species is typically cultivated as a cash crop in northwest China [4] and is an ideal genetic resource for identifying abiotic stress tolerance genes. Given its high resilience to adverse conditions, research studies have focused on identifying saline–alkali- and drought-tolerant genes from *S. alopecuroides* and utilizing them in the development of novel saline–alkali-tolerant crop germplasm. Indeed, previous studies have already identified several saline–alkali tolerance genes from *S. alopecuroides* and reported the underlying mechanism of its salt tolerance [1,6]. A CHY-zinc finger and RING finger (CHYR) family gene was reportedly found during the analysis of saline–alkali and drought tolerance genes in *S. alopecuroides*; however, its role in plant saline–alkali and drought tolerance remains elusive.

The CHYR proteins belong to the RING-type E3 ubiquitin ligase family and contain the CHY-zinc finger and RING-H2 finger domains [7]. The N-terminal of the CHY-zinc finger domain contains conserved “CxHY” amino acid sequences, including His and Cys residues, which are involved in protein interactions, ubiquitination, and zinc ion binding [8,9,10,11]. The RING-H2 finger domain is located on the C-terminus of CHYR, binds two zinc atoms, and may be involved in protein–protein interactions [12], whereas the CHYR N-terminus comprises conserved hemerythrin domains, such as AtBTSL1 and AtBTSL2 [12,13]. *Arabidopsis* BTS plays a vital role in drought stress response by promoting the degradation of the vascular plant one-zinc finger 1/2 (VOZ1/2) protein [14]. Tomato SlVOZ1 is mainly localized in the cytoplasm, but abscisic acid (ABA) can promote the migration of its phosphorylated form to the nucleus; functional analysis has demonstrated that SlVOZ1 plays a positive regulatory role in tomato flowering and drought response [15]. A total of 16 genes encoding GmCHYRs with conserved CHY-zinc finger and RING finger domains have been identified in soybean and are divided into 3 subfamilies [7], of which members of subfamilies I and II are primarily involved in saline–alkali stress, whereas those of subfamily III are induced or inhibited by dehydration and saline–alkali stress [7]. Moreover, 18 *TaCHYR* genes have been identified in wheat. TaCHYRs are upregulated under drought stress and have different expression patterns under salt, cold, and high-temperature stress [16].

CHYR regulates plant growth, development, and stress response through ubiquitination and degradation [11,17,18]. The specificity of the ubiquitination substrate is primarily determined by several E3 ubiquitin ligases [19,20]. E3 ubiquitin ligases containing CHY-zinc finger domains accumulate in the cytoplasm or nucleus and play a pivotal role in mediating the abiotic stress responses through the ABA, ethylene, and mitogen-activated protein kinase signaling pathways [21]. CHYR1, an *Arabidopsis* E3 ubiquitin ligase, is phosphorylated upon interaction with SnRK2.6, a critical protein kinase in the ABA signal transduction pathway. The overexpression of phosphorylated CHYR1 promotes stomatal closure and reactive oxygen species production and improves drought resistance in plants [11].

With climate change further exacerbating the abiotic stress faced by plants, understanding the environmental adaptability characteristics of *S. alopecuroides* becomes crucial given its various medicinal properties. Consequently, research has been increasingly focusing on investigating related genetic resources and the mechanisms underlying plant stress resistance in plants [1]. Nonetheless, studies on the stress resistance of *S. alopecuroides* remain limited, mainly focusing on its molecular biology aspects. Therefore, the present study aimed to identify and analyze the *CHYR* family genes in *S. alopecuroides* and investigate the role of *SaCHYR* family genes in its response to salt, alkali, and drought stress using expression-level analyses of *S. alopecuroides*. Our findings demonstrated the effect of *SaCHYR4* overexpression on enhancing stress resistance in *Arabidopsis thaliana*, providing insights into the stress resistance of *S. alopecuroides* and novel genetic resources for breeding stress-resistant crops.

## 2. Results

### 2.1. Identification and Classification of CHYR Genes in S. alopecuroides

A total of 18 *CHYR* family genes were identified in *S. alopecuroides*. Based on the phylogenetic analysis of the *SaCHYR* genes and their chromosomal locations, they are referred to as *SaCHYR1* to *SaCHYR18*. The predicted protein lengths of the SaCHYRs ranged from 259 to 1287 amino acids, their estimated molecular weights ranged between 29.6 and 148.6 kDa (Table 1), and their isoelectric point values were between 5.52 and 7.83. Subcellular localization prediction showed that the SaCHYR proteins were mainly localized in the nucleus and chloroplast, but SaCHYR14 was predicted to be localized in extracellular space.

To better understand the evolutionary relationships between and the classification of SaCHYRs, we generated phylogenetic trees based on the amino acid sequences of 41 CHYRs from 3 species belonging to the identified *CHYR* gene family, namely, *Arabidopsis*, soybean, and *S. alopecuroides* (Figure 1 and Appendix A). The SaCHYRs could be divided into three clades: *SaCHYR1*, *SaCHYR2*, *SaCHYR3*, *SaCHYR4*, *SaCHYR12*, and *SaCHYR14* were grouped together with *AtCHYR1* and *GmCHYR1* and belonged to clade I; *SaCHYR7*, *SaCHYR8*, *SaCHYR9*, *SaCHYR10*, *SaCHYR16*, and *SaCHYR18* belonged to clade II and were homologous to *AtCHYR5* and *GmCHYR2*; and *SaCHYR5*, *SaCHYR6*, *SaCHYR11*, *SaCHYR13*, *SaCHYR15*, and *SaCHYR17* belonged to clade III and were clustered with *AtCHYR4* and *GmCHYR8*. The phylogenetic analysis results showed that the *SaCHYR* genes were paired and evenly distributed in three subfamilies.

### 2.2. SaCHYR Gene Structure and Conserved Motif Analysis

To understand the structural variations and characteristics of the *SaCHYR* genes, we analyzed the exon–intron structures and conserved motifs of these genes (Figure 2a–c). The minimum and maximum numbers of exons on the *SaCHYRs* were 11 and 14, respectively (Figure 2b). We classified the *SaCHYR* genes with similar exon and intron structures into three subfamilies. Of note, most of the more evolutionarily related genes shared similar exon–intron structures.

To analyze the sequence characteristics of the CHYR family members of *S. alopecuroides*, 15 conserved amino acid sequence motifs were predicted (Figure 2c,d). All SaCHYR proteins contained the conserved CHY-zinc finger, zinc ribbon domains, and C3H2C3-type Ring finger (Figure 2c); these results validated the reliability of the *CHYR* family of genes identified in *S. alopecuroides*. Based on the distribution of the conserved domains, the SaCHYR proteins in each group showed similar motif distributions. The SaCHYR protein sequence alignment showed that the CHY-zinc finger and zinc ribbon domains were highly conserved (Appendix A).

In addition, the group III members had two or three heme domains at the N-terminus, consistent with the results obtained in *Arabidopsis*, soybean, and wheat, which may explain the involvement of the CHYR members of group III in the regulation of iron homeostasis (Figure 2c).

We further analyzed the conserved motifs in the *SaCHYR* gene family using MEME and found that the detected motifs 1, 7, and 3 constituted the CHY-zinc finger domain; 5 and 11 constituted the ring finger domain; 4 and 2 constituted the zinc ribbon domain; and 9 and 7 constituted the merythrin domain. Conserved motifs 9 and 13, with similar heme domains, were found in group III members (Figure 2c). Overall, the *SaCHYR* genes of the internal members of each group showed similar motif compositions and combinations.

### 2.3. Chromosomal Mapping of SaCHYR Genes in S. alopecuroides

A chromosomal distribution analysis showed that the 18 identified *SaCHYR* genes were distributed on 12 chromosomes, with 2 genes each on chromosomes 9–14 and 1 gene each on chromosomes 1, 2, and 5–8 (Figure 3). Group I members were mainly distributed on chromosomes 1, 2, 5, 6, and 9–14; group II on chromosomes 9 and 10; and group III on chromosomes 7, 8, and 11–14 (Figure 3). Our results showed no significant correlation between the number of *SaCHYR* genes and chromosome length.

### 2.4. Cis-Regulatory Element Analysis of SaCHYR Gene Promoters

To explore the regulatory mechanisms of the *SaCHYR* gene family in response to abiotic stress, we analyzed the CREs of the *SaCHYR* gene promoter using PlantCARE. We identified 52 CREs in the promoter regions of 18 *S. alopecuroides CHYR* gene family members, including elements related to photoresponsiveness, stress responsiveness, hormone responsiveness, and growth- and development-related elements (Figure 4a,b). The *SaCHYR* promoter included several stress and hormone response elements, especially stress response elements, mainly including the anaerobic response element (ARE) and MBS (MYB binding site; Figure 4c). These results are similar to those predicted based on the CREs of the *CHYR* family gene promoters in soybean and wheat [7,16].

### 2.5. Analysis of the 3D Structure of SaCHYR Proteins

The three-dimensional (3D) protein structure helps to understand protein function; therefore, we analyzed the 3D structure of the SaCHYR protein. SWISS-MODEL was used to predict the 3D structure of the SaCHYR protein, and the quality of the model was further evaluated using QMEANDisCo values. All 18 SaCHYR proteins had QMEANDisCo values of >0.72, indicating the reliability of the predicted 3D structure (Figure 5, Appendix A). The results showed a high similarity between the SaCHYR3/4, SaCHYR5/6, SaCHYR7/9, SaCHYR8/10, SaCHYR11/13, SaCHYR12/14, SaCHYR15/17, and SaCHYR16/18 protein pairs. Slight differences were noted between the SaCHYR1 and SaCHYR2 proteins. Combining the 3D structures of the SaCHYR proteins could facilitate a clearer understanding of the similarities and differences among family members, which would provide a reference for further understanding of the function of SaCHYR proteins in *S. alopecuroides*.

### 2.6. Prediction of Proteins Interacting with S. alopecuroides CHYR

Using the soybean CHYR protein as a reference, the proteins interacting with the CHYR protein in *S. alopecuroides* were predicted to determine the biological function of the CHYR protein in *S. alopecuroides*. The results revealed 31 CHYR-interacting proteins, including methyltransferase, AAA_assoc domain-containing protein, UmuC domain-containing protein, DNA repair protein REV1, ubiquitin-like domain-containing protein, AAA domain-containing protein, DNA repair protein rev1, ubiquitin-like domain-containing protein, AAA domain-containing protein, RING-type domain-containing protein, Tify domain-containing protein, and cytochrome P450 (Figure 6). Such results provide a reference for the functional study of the *CHYR* family genes in *S. alopecuroides*, which were predicted to determine the biological function of the CHYR protein in *S. alopecuroides*.

### 2.7. Expression Patterns of SaCHYR Genes Based on RNA-Seq Analysis

To explore the expression levels of the *CHYR* family genes in different *S. alopecuroides* tissues, transcriptome sequencing was performed on the roots, stems, and leaves, and the expression levels of the SaCHYRs in various tissues were determined (Figure 7a). *SaCHYR1*, *SaCHYR2*, *SaCHYR4*, *SaCHYR8*, *SaCHYR12*, *SaCHYR14*, and *SaCHYR16* were expressed at higher levels in the roots than in the stems and leaves; *SaCHYR15*, *SaCHYR17*, and *SaCHYR18* were highly expressed in the stems; and *SaCHYR5*, *SaCHYR6*, and *SaCHYR11* were expressed mainly in the leaves. These results demonstrated the spatial-specific expression of the *CHYR* genes in *S. alopecuroides*.

To explore the role of the *SaCHYR* genes in response to abiotic stress, we exposed *S. alopecuroides* to salt, alkali, and drought stress. Transcriptome sequencing was performed on the root tissues to explore the expression levels of the 18 *SaCHYR* genes post-treatment. The results showed that *SaCHYR4*, *SaCHYR5*, *SaCHYR11*, and *SaCHYR14* were induced significantly by alkali stress, *SaCHYR6* was induced significantly by salt stress, and *SaCHYR1*, *SaCHYR2*, *SaCHYR9*, *SaCHYR12*, and *SaCHYR17* were induced significantly by drought stress (Figure 7b). According to the three clades of the *CHYR* family in *S. alopecuroides*, we found that the *SaCHYR* genes in clade I were mainly involved in the drought stress response and those in clade III in the alkali stress response.

A quantitative analysis of the *CHYR* family genes in *S. alopecuroides* under alkali and drought stress revealed that the expression trends of eight *SaCHYR* genes were consistent with those identified via transcriptomic quantitative analysis, ultimately confirming the reliability of our transcriptomic data (Figure 7). Based on the quantitative levels under alkali and drought stress for multiple periods, the expression of *SaCHYR1* under alkali and drought stress showed opposite trends (Figure 7c); *SaCHYR4* was initially upregulated under alkali stress and then consistently downregulated under drought stress (Figure 7d). Of note, *SaCHYR8* was more significantly responsive to drought stress (Figure 7e). *SaCHYR9* showed no significant change under alkali stress but was upregulated under drought stress (Figure 7f). *SaCHYR11* showed the same expression trend under alkali and drought stress (Figure 7g), while *SaCHYR14* showed the highest expression 48 h after alkali and drought stress (Figure 7h). *SaCHYR16* was significantly downregulated under alkali and drought stress (Figure 7i), while the expression of *SaCHYR17* remained unaltered under alkali and drought stress conditions, but relatively increased under drought stress (Figure 7i). In summary, the expression patterns of the *CHYR* genes in *S. alopecuroides* in response to abiotic stress exhibited considerable variation, highlighting the diverse and intricate roles played by the *SaCHYR* genes in the plant response to abiotic stress.

We further investigated the *SaCHYR4* expression patterns under alkali and drought stress at different time points, showing that alkali stress led to the highest *SaCHYR4* expression in the roots after 12 h, while drought stress triggered the highest expression in the roots and leaf tissues after 24 h (Appendix A).

### 2.8. SaCHYR4 Overexpression Increased Alkaline Tolerance in A. thaliana

To verify the function of the *CHYR* family genes of *S. alopecuroides* in the abiotic stress response, we overexpressed *SaCHYR4* in *A. thaliana*. The wild-type (WT) and overexpressed strains (OX, including OE5 and OE21) plants at the germination stage were subjected to salt, alkali, and drought stress. The results showed that *SaCHYR4* overexpression increased the germination and green rates of cotyledons in *A. thaliana* under alkali and drought stress (Appendix A). No significant changes were observed between the OX and WT strains under salt stress. Furthermore, to determine the effects of *SaCHYR4* overexpression on the alkali and drought tolerance of *A. thaliana*, we performed stress treatment and physiological indicator detection at the seedling stage.

In the present study, different NaHCO_3_ concentrations were used to simulate alkali stress treatments; no significant changes were observed between the OX and WT plants under normal conditions (Appendix A). However, the OX plants germinated significantly earlier than the WT plants, and their root lengths were significantly longer than those of the WT plants under alkaline stress (Appendix A). Seedlings were treated with 300 mM NaHCO_3_-simulated alkali stress, showing that the growth of all strains was similar under normal conditions (Figure 8a). After 8 days of alkali stress exposure, the wilting and yellowing of the WT leaves were more severe than those of the OX leaves, and OX seedlings were less damaged with better growth states (Figure 8a). The survival rates of the OX plants were significantly higher than those of the WT plants under alkaline stress (Figure 8b). The fresh weight, chlorophyll, and malondialdehyde contents in the OX plants were significantly higher, and the malondialdehyde content was significantly lower (reduced by 37–50%) than the levels in the WT plants under alkaline stress (Figure 8c–e). Overall, *SaCHYR4* overexpression significantly improved the alkaline tolerance of *A. thaliana*.

### 2.9. SaCHYR4 Overexpression Increased Drought Tolerance in A. thaliana

We simulated drought stress using different concentrations of mannitol to treat the OX and WT plants. The germination rate of the OX plants under drought stress did not significantly differ from that of the WT plants; however, the green rate of cotyledons was substantially higher in the OX plants than in the WT plants (Appendix A). The root length of the OX plants significantly exceeded that of the WT plants under drought stress, indicating that *SaCHYR4* overexpression improved the drought tolerance of *A. thaliana* (Appendix A).

The drought tolerance of the OX plants was determined through water control at the seedling stage. With the extension of drought stress, the wilting and shrinking of the WT leaves were more severe relative to the OX leaves (Figure 9a). After 4 days of rehydration, the OX leaves recovered quickly, whereas most WT plants did not recover, and nearly all were dry (Figure 9a). After drought stress, the survival rates of OX-5 and OX-21 were significantly higher by 78% in the OX plants than in the WT plants, the chlorophyll content was significantly higher in the OX plants than in the WT plants, and the malondialdehyde content was lower in the OX plants than in the WT plants (Figure 9b–d). The water loss rate of the OX plants under drought stress was significantly lower by 5–6% than that of the WT plants, suggesting that *SaCHYR4* improved the drought tolerance of OX by regulating the water loss of *A. thaliana* under drought stress (Figure 9e). Overall, our results showed that *SaCHYR4* overexpression in *A. thaliana* can enhance the drought stress tolerance of OX plants.

### 2.10. SaCHYR4 Overexpression Alters the Transcription Levels of Stress Response Genes

To further explore the effect of *SaCHYR4* overexpression on the stress response in *Arabidopsis*, the expression levels of the stress response genes in *Arabidopsis* were quantitatively analyzed using qRT-PCR. The expression levels of *SOD*, *APX1*, *APX2*, *POD70*, *POD72*, *PDF1.2*, *RD17*, *RD29A*, and *RD29B* in the *SaCHYR4* overexpression lines were significantly higher than those in the WT *Arabidopsis* under alkali and drought stress conditions (Figure 10). This suggests that *SaCHYR4* overexpression may respond to alkali and drought stress by regulating the expression of stress response genes. Additionally, the expression levels of *RD17* and *RD29A* were significantly higher in *SaCHYR4* overexpression lines than in the WT under control conditions, and the expression dynamics of each stress response gene were different under alkali and drought stress conditions, indicating some differences in the response to alkali and drought stresses and that *SaCHYR4* plays an important role in plant abiotic stress response.

## 3. Discussion

The *CHYR* gene family member *SaCHYR4*, which was selected from *S. alopecuroides* through stress resistance screening, showed a positive response to abiotic stress [1]. To further explore the genetic information of the *CHYR* family genes in *S. alopecuroides* and their roles in the plant response to abiotic stress, we analyzed the whole genome of *S. alopecuroides*. To the best of our knowledge, this is the first study to identify and classify 18 *CHYR* family genes and validate the function of *SaCHYR4*. Moreover, the results were verified in transgenic *Arabidopsis SaCHYR4* under alkali and drought stress. Our results provide a theoretical reference for the improvement and application of *CHYR* family genes and a basis for further exploration of the response of *SaCHYR* genes to abiotic stress, which is of great significance for elucidating the anti-stress mechanism of *S. alopecuroides*.

In this study, 18 genes of the *SaCHYR* family were identified based on the CHY-zinc finger, RING-H2 finger, and zinc-ribbon domains of their proteins in *S. alopecuroides*, consistent with the identification of *CHYR* family genes in other species. To date, 16 *CHYR* family genes have been identified in soybean [7], 7 in *Arabidopsis*, and 18 in wheat [16], suggesting no significant differences in the numbers of *CHYR* family genes between dicotyledonous and monocotyledonous plants, which might be attributable to similar genome sizes. Moreover, the gene structure of the *SaCHYR* family in *S. alopecuroides* was similar to that of the soybean *CHYR* family, with exon–intron numbers ranging between 10 and 15.

A conserved domain analysis showed a high similarity between SaCHYR family proteins and GmCHYR, and phylogenetic tree clustering showed a consistent gene structure and motif distribution among proteins in the same branch. In this study, we predicted 15 motifs in SaCHYRs, consistent with those for *Arabidopsis* and soybean [7]. Overall, the CHYR family proteins seemed to be conserved to some extent among species, suggesting that they play crucial roles in plant growth, development, and adaptation to adverse environmental conditions.

The CHYR protein, a Ring-type E3 ubiquitin ligase, participates in regulating plant growth, development, and environmental adaptation through ubiquitination. The expression of all *TaCHYRs* identified in wheat was induced by drought stress, showing similar expression patterns [16]. In this research, the expression of 14 of the 18 *CHYR* family genes identified in *S. alopecuroides* was significantly triggered by salt, alkali, and drought stress. Ring-type E3 ubiquitin ligases contain the RING zinc-finger domain, which plays a key role in abiotic stress responses through the ABA signaling pathway. The *Arabidopsis* RING finger E3 ubiquitin ligase XBAT35 regulates the accumulation of the ABA receptor PYL4 by inducing the ubiquitination-mediated degradation of VPS23A, thus playing a role in ABA signaling [20]. Moreover, XBAT35.2 and the deubiquitinating enzymes UBP12/13 play the roles of “accelerator” and “brake”, finely regulating the ubiquitination level of VPS23A and affecting the level of the ABA receptor protein in response to external environmental changes [22]. Ubiquitin ligase XBAT35.2 promotes pathogen defense by mediating ACD11 degradation, but weakens plant responses to abiotic stress, suggesting that XBAT35.2 has opposite effects on abiotic and biotic stress [23]. *Arabidopsis* E3 ubiquitin ligases AIRP3 and UBC27 can interact with the ABA co-receptor ABI1; the function of AIRP3 relies on UBC27 and collaboratively affects the stability of ABI1 to regulate ABA signaling and the mechanism underlying the plant drought stress response [24]. The Ring-type E3 ubiquitin ligase CaAIRF1 interacts with and ubiquitinates class A PP2Cs (CaADIP1 and CaAIPP1), ultimately regulating their stability and mediating ABA signal transduction in plants [25,26]. Here, we found that almost all *SaCHYR* genes contained CAACTG, a binding site (MBS) of the MYB transcription factor responding to drought stress, and ABRE (ACGTG)-responding elements; this suggests that *SaCHYR* genes regulate abiotic stress through ABA signaling pathways. The Rice Ring-type E3 ubiquitin ligase OsHTAS controls H_2_O_2_ homeostasis through ABA- and DST-dependent pathways, thereby regulating heat tolerance in rice [27]. *OsRZF34*, a rice gene homologous to *Arabidopsis AtCHYR1*, increases stomatal opening and transpiration rate when overexpressed in *Arabidopsis* and rice, thereby enabling the faster cooling of leaves [28]. *Populus euphratica* overexpressing *PeCHYR1* showed reduced stomatal opening due to an increased hydrogen peroxide (H_2_O_2_) content. Transgenic poplars showed a high sensitivity to exogenous ABA and drought tolerance [29]. Moreover, the Rice Ring-type E3 ligase OsRINGzf1 enhances the water retention capacity of rice by promoting OsPIP2;1 degradation, thus positively regulating drought resistance [30]. Our results showed that *SaCHYR4* overexpression in *Arabidopsis* significantly improved drought tolerance; the water loss rate of OX was considerably lower than that of WT, suggesting that *SaCHYR4* overexpression enhances drought tolerance by increasing the water retention rate of *Arabidopsis*.

Ring-type E3 ubiquitin ligases not only regulate enzyme activity via ubiquitination, but also the expression of transcription factors (for example, WRKY and MYB) to regulate stress response in plants. CHYR1 controls the binding of WRKY70 mediated by the phosphorylation of various cis-acting elements of SARD1, thereby regulating homeostasis between immunity and growth [31]; this implies that the E3 ubiquitin ligase functions by regulating the stability of its target proteins, and its function is regulated by protein phosphorylation and dephosphorylation. The RING-HC E3 ubiquitin-protein ligase SR1 regulates the expression of RAP2.2, a member of ERF-VII, by regulating the stability of the transcription factor WRKY33, thereby affecting plant hypoxia signal transduction in response to plant tolerance to flooding stress [32]. The *Arabidopsis* ubiquitin E3 ligases MIEL1 and MYB30 co-regulate the key ABA signaling pathway transcription factor ABI5, which is involved in seed germination [33]. The Ring-type E3 ligase MdMIEL1 in apples can mediate the ubiquitination and degradation of MdBBX7 through the 26S proteasome pathway, negatively regulating the response to drought stress [34]. MdSINA3 inhibits the promoting effects of MdBBX37 on leaf senescence by regulating MdBBX37 degradation. MdBBX37 interacts with MdABI5 in ABA signaling and MdEIL1 in ethylene signaling, affecting ABA- and ethene-mediated leaf senescence, respectively [35]. In this study, we predicted 31 proteins interacting with the SaCHYR protein, including methyltransferase, ubiquitin-like domain-containing protein, and cytochrome P450. Methyltransferase and RING-type domain-containing proteins are reportedly involved in the regulation of plant responses to saline–alkali stress [25]. Hence, Ring-type E3 ubiquitin ligases of the CHYR subclass may be involved in regulating plant responses to environmental stress through multiple pathways, further indicating the functional diversity of CHYR family proteins.

Currently, there is limited research on the alkali stress resistance of CHYR family proteins. In this study, we utilized RNA-seq data to demonstrate that alkaline and drought stress strongly induce the expression of the *SaCHYR4* gene in *S. alopecuroides*. As the genetic transformation system of *S. alopecuroides* had not been previously established, we examined the function of *SaCHYR4* through heterologous expression in *Arabidopsis*. *SaCHYR4* overexpression significantly enhanced the alkaline stress tolerance of *Arabidopsis*; however, there was no substantial change in the salt stress tolerance of OX plants, implying potential differences in the plant responses to salt and alkali stress. Additionally, the SaCHYR4 protein exhibits unique characteristics in regulating plant stress resistance. The *SaCHYR* family genes play a crucial role in modulating plant responses to abiotic stresses (such as salt, alkali, and drought). The overexpression of *SaCHYR4* in *A. thaliana* significantly improved the alkaline and drought tolerance in seedlings. The expression levels of *SOD*, *APX1*, *APX2*, *POD70*, *POD72*, *PDF1.2*, *RD17*, *RD29A*, and *RD29B* in the SaCHYR4 overexpression lines were significantly changed under alkali and drought stress. These findings provide a foundation for studying the role of the *SaCHYR* family genes in crop stress resistance.

CHYR, a key enzyme in the ubiquitination pathway, considerably influences plant responses to abiotic stress. However, to date, studies on the function of the CHYR protein have primarily focused on model plants, such as *Arabidopsis* and rice. Therefore, it is vital to explore the mechanism of CHYR resistance in resistant plants. Understanding the anti-stress mechanism of *S. alopecuroides* is crucial, as it serves as a valuable resource for stress resistance.

This study has certain limitations. The specific mechanism of action of the *SaCHYR4* gene in the resistance of *S. alopecuroides* was not thoroughly explored, despite the identification of the *CHYR* gene family. The role of its analog in the resistance mechanism of *Arabidopsis* also remains unclear. Considering the complexity of plant responses to environmental stress and the immaturity of the genetic transformation system of *S. alopecuroides*, studying the function of *SaCHYR* genes poses significant challenges. Future research should focus on exploring the resistance-related genes of *S. alopecuroides*, analyzing the response mechanism of each gene in the *SaCHYR* family to abiotic stress, and investigating the relationship between its medicinal use and resistance. Furthermore, understanding the molecular characteristics of *S. alopecuroides* will help to maximize its economic value.

## 4. Materials and Methods

### 4.1. Plant Materials

*Sophora alopecuroides* seeds were collected from the Korla region of Xinjiang (N 41.81°, E 86.19°). Full seeds were then selected and placed in triangular bottles; 98% sulfuric acid (H_2_SO_4_) was slowly added to soak the seeds at room temperature (18–25 °C) for 10 min, during which, the seeds were slowly shaken. Distilled water was then added four or five times to remove residual H_2_SO_4_ from the seed coat. The seeds were soaked for 2 days, and hilum was uncovered and sowed (peat soil:vermiculite = 3:1) under the following culture conditions: day/night 16 h/8 h, temperature 26/22 °C, humidity 65%, light intensity 30,000 lx, Changchun, Jilin, China. At 4 weeks of age, the *S. alopecuroides* seedlings were selected, grouped, and exposed to 150 mM NaHCO_3_, 200 mM NaCl, and 8% PEG6000 for 72 h [1]. Fresh root, stem, and leaf tissues were collected from each treatment, frozen, and stored at −80 °C for later use.

### 4.2. Transcriptome Sequencing and Analysis

RNA was extracted from the root, stem, and leaf tissues of *S. alopecuroides* using TRIzol [1,36]. A library was constructed using the chain-specific construction method, and the insert size of the library was assessed using an Agilent 2100 bioanalyzer to determine the library quality. Sequencing was then performed using PacBio sequencing technology (BGI Genomics Co., Ltd., Wuhan, China). Clean reads generated by an NGS platform were mapped to the genome sequence of *S. alopecuroides* using HISAT2 v2.2.1 with default parameters [37]. Gene expression (expressed in transcripts per million, TPM) was calculated using StringTie v2.20 [37].

### 4.3. Genome-Wide Identification of the CHYR Gene Family of S. alopecuroides

An *S. alopecuroides* genome database was applied to identify the *CHYR* family genes. A total of 23 known AtCHYR and GmCHYR protein sequences were collected from The Arabidopsis Information Resource and Soybase. These protein sequences were used as queries to perform BLAST searches against the *S. alopecuroides* genome database to identify the *CHYR* genes. The CHY-zinc finger (PF05495), ring-finger domain (PF13639), and zinc-ribbon domain (PF14599) from Pfam (http://pfam.xfam.org, accessed on 25 September 2023) were used to perform HMM searches with HMMER (http://hmmer.org/download.html, accessed on 25 September 2023) and to identify *SaCHYR* genes [38]. The results from two different methods were verified by analyzing the intact gene domain using Pfam, SMART, and the Conserved Domain Database. Subcellular SaCHYR localization was predicted using Cell-2.0 (http://cello.life.nctu.edu.tw/, accessed on 25 September 2023).

### 4.4. Multiple Sequence Alignments and Phylogenetic Tree Construction

Multiple sequence alignments were performed for known AtCHYR and GmCHYR proteins, and the full-length SaCHYR protein sequences were predicted using clustalW (https://www.genome.jp/tools-bin/clustalw, accessed on 27 September 2023). A phylogenetic tree was constructed based on 1000 bootstrap samples via the maximum likelihood method using MEGA X [39], and evolview was used to visualize the phylogenetic trees (https://www.evolgenius.info/evolview/, accessed on 29 September 2023) [40].

### 4.5. Chromosomal Location, Gene Structure, Conserved Domain, and Cis-Regulatory Element Analyses

The chromosomal locations of the *SaCHYR* family genes were identified based on the genome of *S. alopecuroides* using TBtools, and all *SaCHYR* family genes were mapped to chromosomes based on their chromosomal positions using TBtools [41]. The genes were named according to their chromosomal positions. TBtools [41] was used to generate the gene structure map of the gene family. The *SaCHYR* family genetic structure, conservative structure domain analysis, and visualization were also performed (https://www.ncbi.nlm.nih.gov/Structure/bwrpsb/bwrpsb.cgi, accessed on 2 October 2023). Based on the genome annotation file, TBtools was used to map the exon–intron composition and location of *SaCHYR*. Conserved motifs in the SaCHYR protein sequences were identified using the MEME program, with 15 being the maximum number of motifs [42]. Protein structure prediction of the *SaCHYR* genes was performed using the NCBI conserved domains (https://www.ncbi.nlm.nih.gov/Structure/cdd/wrpsb.cgi, accessed on 2 October 2023). A cis-regulatory element (CRE) analysis of 18 *SaCHYR* genes’ promoter regions (2000 bp) was performed using the PlantCARE database (http://bioinformatics.psb.ugent.be/webtools/plantcare/html, accessed on 3 October 2023).

### 4.6. Homology Modeling of 3D SaCHYR Protein Structures

The secondary structure of the SaCHYR protein was predicted using the online tool SOPMA (https://npsa-prabi.ibcp.fr/cgi-bin/npsa_automat.pl?page=npsa%20_sopma.html, accessed on 4 October 2023). The tertiary structure was modeled and visualized using the SWISS-MODEL tool (https://swissmodel.expasy.org/interactive, accessed on 4 October 2023).

### 4.7. Construction of the Protein–Protein Interaction Network

Protein–protein interactions were predicted using the protein sequences of the *S. alopecuroides CHYR* family members via https://cn.string-db.org/cgi/input.pl, accessed on 20 December 2023. Soybean was selected as the reference plant. The minimum required interaction score was set to 0.400; active interaction sources, including text mining, experiments, databases, co-expression, neighborhood, gene fusion, and co-occurrence, were used to construct a CHYR protein interaction network.

### 4.8. Analysis of the Pattern of SaCHYR Family Gene Expression in Response to Abiotic Stress

Using transcriptome data, we analyzed the expression patterns of the *SaCHYR* gene family in different tissues of *S. alopecuroides* and assessed the changes in the expressions of these genes under salt, alkali, and drought stress conditions. To further validate the role of the *CHYR* genes in *S. alopecuroides* in response to abiotic stress, the expression dynamics of 8 *SaCHYR* genes were detected using RT-qPCR at different time points after alkali and drought stress exposure. The total RNA was extracted from the root tissues of 150 mM NaHCO_3_- and 8% PEG-treated *S. alopecuroides* using TriZol (Genestar, P118-05) (Chiyoda-ku, Tokyo), and reverse transcription was performed using a commercial kit (Genestar, A232-10). After determining the primer specificity using TBtools, RT-qPCR was performed using SGExcel FastSYBR Mixture (Sangon Biotech, B532955) (Shanghai, China) and CFX connect (Bio-Rad) (Hercules, CA, USA). The experiment was performed at 95 °C for 30 s, 95 °C for 5 s, 60 °C for 30 s for 40 cycles, 95 °C for 10 s, and 65 °C for 5 s (with a 0.5 °C gradient increase to 95 °C). Actin served as the internal reference gene, and the relative gene expression was analyzed using the 2^−∆∆CT^ method, with three biological replicates per sample; each experiment was repeated three times (primers are listed in Appendix A).

### 4.9. Cloning of the SaCHYR4 Gene

Based on the obtained *SaCHYR4* nucleic acid sequence, primers were designed with Primer 6.0 using *S. alopecuroides* cDNA as a template and a high-fidelity enzyme for *SaCHYR4* cloning (Appendix A). The recovered PCR products were connected to the cloning vector pMD18-T for amplification, and the obtained sequencing results were confirmed to be consistent with the transcriptome sequencing results and used to construct the expression vector.

### 4.10. Plasmid Construction and Genetic Transformation of A. thaliana

The *SaCHYR4* coding sequence was recombined into the plant expression vector pCHF3300 via double-enzyme digestion to construct the expression vector pCHF3300-*SaCHYR4*. Wild *A. thaliana* plants were subjected to *Agrobacterium* EHA105-mediated infection. Seeds were harvested using Basta for screening, and PCR was used to detect positive strains. Seeds were harvested from the T_2_ generation for further experiments.

### 4.11. Functional Analysis of SaCHYR4 in A. thaliana

*Arabidopsis* seeds were disinfected, suspended in 0.1% AGAR solution, planted in 1/2 MS medium, vernalized at 4 °C for 3 days, and placed in a light incubator (day/night 16/8 h, temperature 24/20 °C, humidity 65%, light intensity 30,000 lx). The germination rate was determined after 3 days, whereas the green seedling rate was determined after 5 days. The germination stage was subjected to salt (1/2 MS medium containing 100 mM NaCl), alkaline (1/2 MS medium containing 7, 8, or 9 mM NaHCO_3_), and drought stress (1/2 MS medium containing 100 or 200 mM mannitol). *Arabidopsis* was cultured in 1/2 MS medium, and when the four leaves were unfurled, they were transferred to high-temperature sterilized soil for culturing (turf soil:vermiculite = 3:1; soil moisture content was the same). The seedlings were exposed to stress after 2 weeks of cultivation, with 300 mM NaHCO_3_ added to simulate alkali stress. Drought stress was applied for 10 days, followed by rehydration. The phenotypes were observed and photographed 2 days later. After stress treatment, the survival rates and leaf water loss rates of the *Arabidopsis* seedlings were evaluated, and the chlorophyll and malondialdehyde content was measured. The selected *Arabidopsis* stress response genes included *SOD*, *APX1*, *APX2*, *POD70*, *POD72*, *FSD2*, *PDF1.2*, *RD17*, *RD29A*, and *RD29B*. The NCBI web site (https://www.ncbi.nlm.nih.gov/tools/primer-blast/index.cgi?LINK_LOC=BlastHome, accessed on 22 March 2024) was used to design the RT-qPCR primers, which were synthesized by Comate Bioscience Co., Ltd. (Changchun, China), Jilin Province. The specific methods have been described previously [43].

### 4.12. Statistical Analysis

One-way analysis of variance was used to analyze differences among the experimental groups. IBM SPSS 19.0 statistical software (IBM, Armonk, NY, USA) and Microsoft Excel 2010 were used for statistical analysis, and GraphPad Prism 8.0 (GraphPad Software, La Jolla, CA, USA) was used for visualization.

## 5. Conclusions

Using genome-wide data, this study identified 18 *S. alopecuroides CHYR* family genes and classified them into 3 subfamilies using a phylogenetic analysis. An analysis of the *CHYR* gene structure, conserved domains, motifs, chromosome distribution, cis-regulatory elements, and 3D protein structure revealed that the CHYR family proteins were relatively conserved. Moreover, an analysis of the expression patterns of the *SaCHYR* family genes indicated their positive response to abiotic stress in *S. alopecuroides*. Notably, *SaCHYR4* overexpression significantly increased *A. thaliana* tolerance to alkaline and drought stress. These findings provide a basis for future studies on the stress resistance mechanism of *S. alopecuroides* and serve as a new genetic resource for breeding stress-resistant crops.

## Figures and Tables

**Figure 1 ijms-25-06173-f001:**
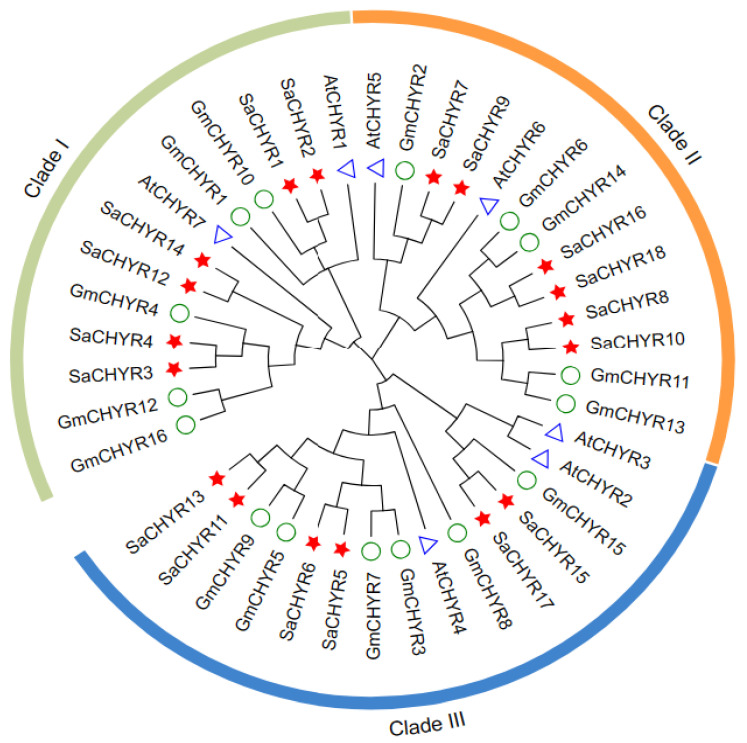
Phylogenetic analysis of CHYR proteins from *Sophora alopecuroides*, *Arabidopsis thaliana*, and *Glycine max* (L.). Circles, triangles, and stars indicate soybean proteins, those of *A. thaliana*, and those of *S. alopecuroides*, respectively. Clades I, II, and III are represented by green, orange, and blue, respectively.

**Figure 2 ijms-25-06173-f002:**
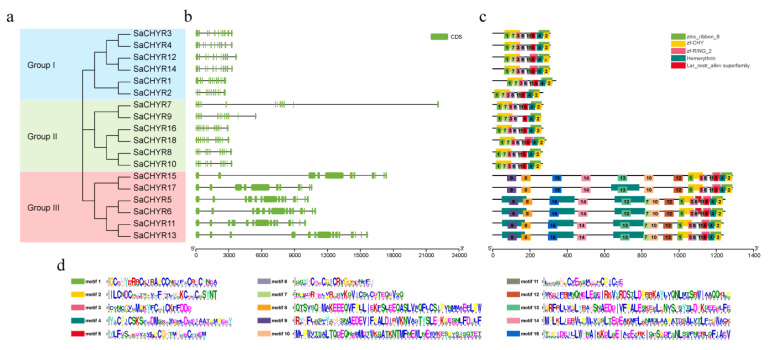
Characterization of *Sophora alopecuroides CHYR* genes. (**a**) Phylogenetic analysis of SaCHYR proteins from *S. alopecuroides.* (**b**) Exon–intron structural features of 18 *SaCHYR* genes. (**c**) Conserved motifs of the amino acid sequences of SaCHYR proteins. (**d**) Amino acid preference analysis of motifs among SaCHYRs.

**Figure 3 ijms-25-06173-f003:**
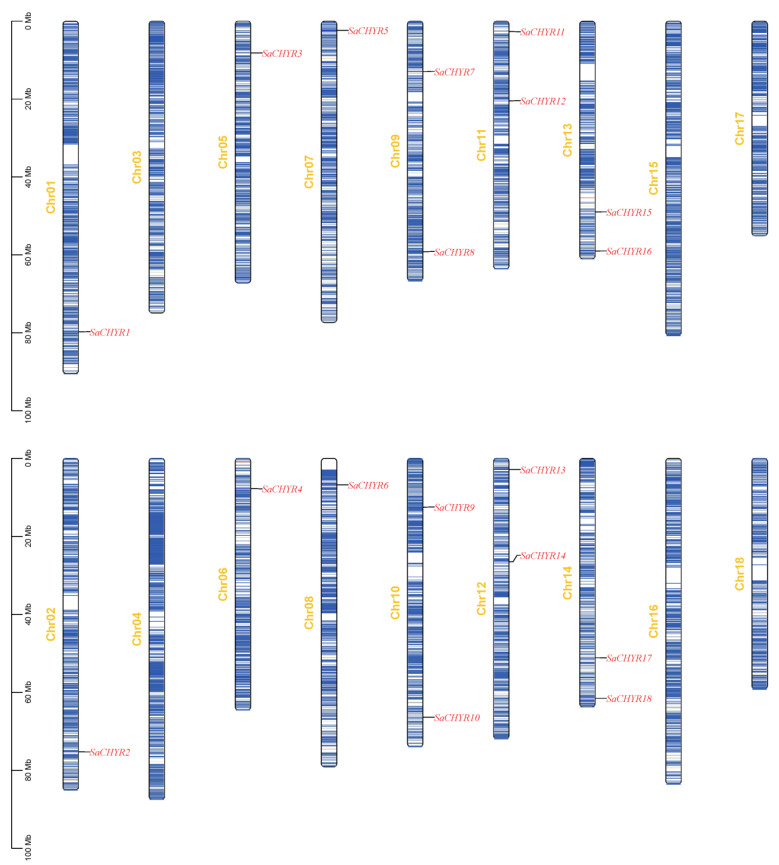
Chromosomal location of *SaCHYR* genes in *Sophora alopecuroides*.

**Figure 4 ijms-25-06173-f004:**
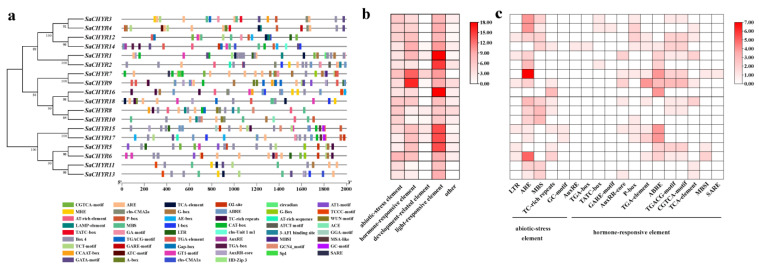
Cis-acting element analysis of *SaCHYR* promoters. (**a**) Predicted promoter cis-acting elements. (**b**) Cis-acting elements in the *SaCHYR* promoters were divided into five types, and their numbers are shown. (**c**) Numbers of environmental stress-related factors and hormone response factors.

**Figure 5 ijms-25-06173-f005:**
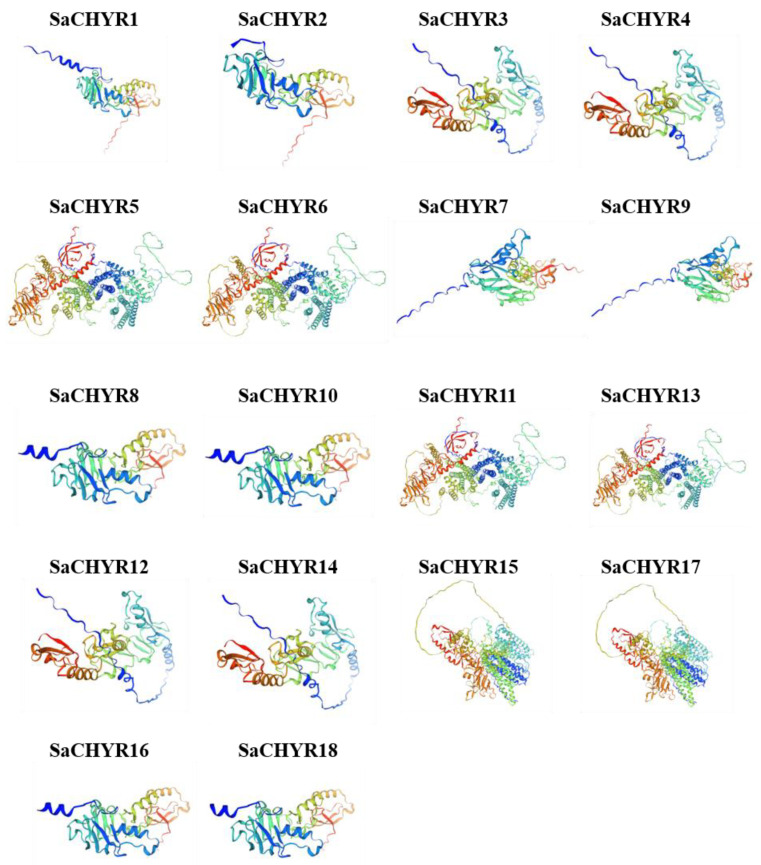
Three-dimensional structural models of 18 SaCHYR proteins.

**Figure 6 ijms-25-06173-f006:**
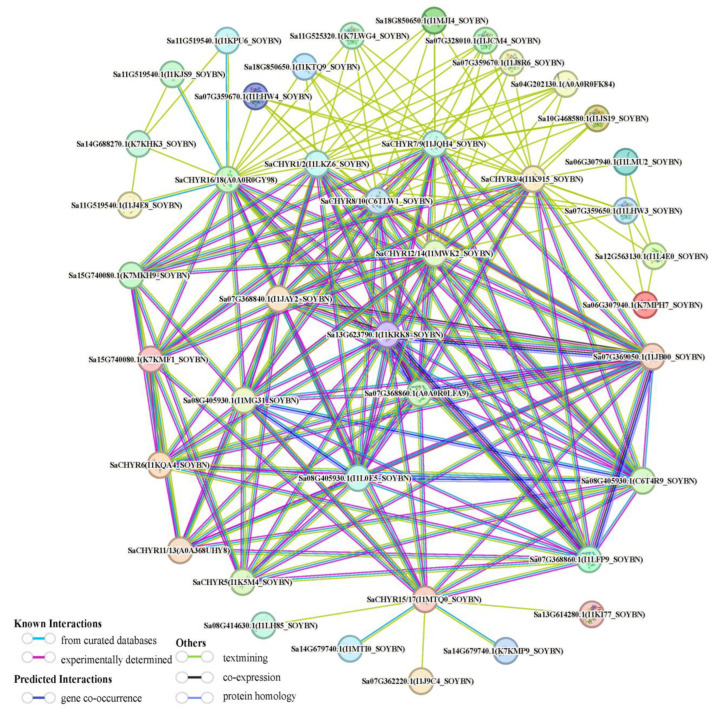
Prediction of SaCHYR-interacting protein networks in *S. alopecuroides*.

**Figure 7 ijms-25-06173-f007:**
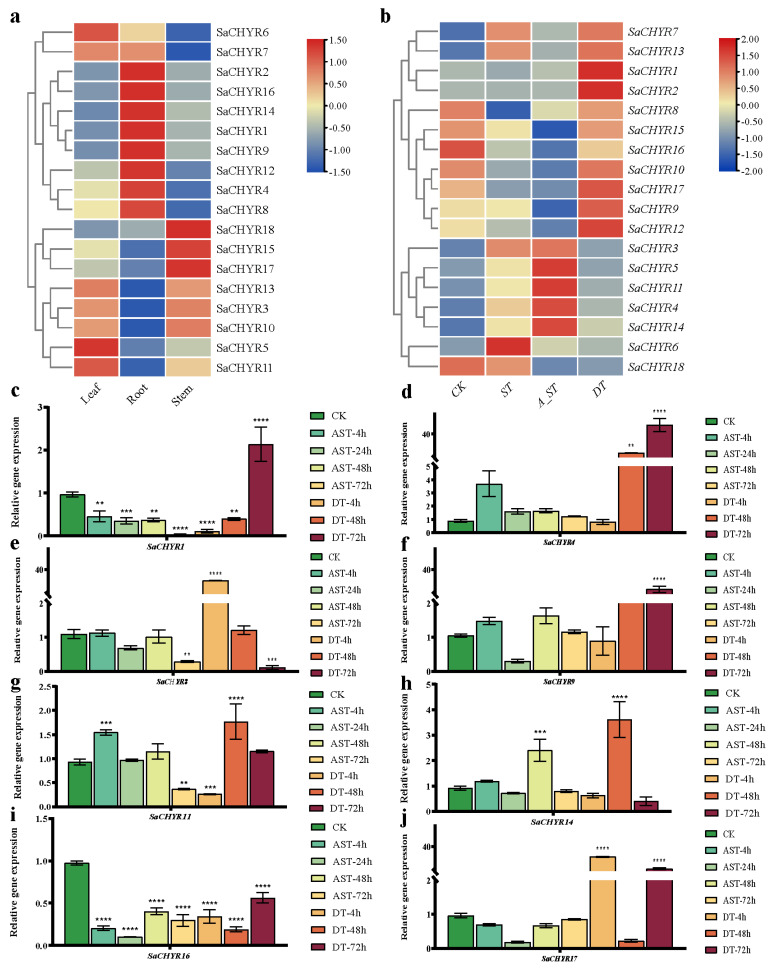
Expression profiles of *CHYR* genes in *Sophora alopecuroides.* (**a**) Expression patterns in leaf, root, and stem tissues. (**b**) Expression patterns under control (CK), salt (ST), alkali (A_ST), and drought (DT) conditions. Expression levels are indicated by the color scale. (**c**–**j**) Relative expression levels of *SaCHYRs* in *S. alopecuroides* at different periods under alkali and drought stress. Data represent means ± SD, with three biological replicates. GraphPad was used to calculate significance using one-way ANOVA. **, *p* < 0.01, ***, *p* < 0.001, ****, *p* < 0.0001.

**Figure 8 ijms-25-06173-f008:**
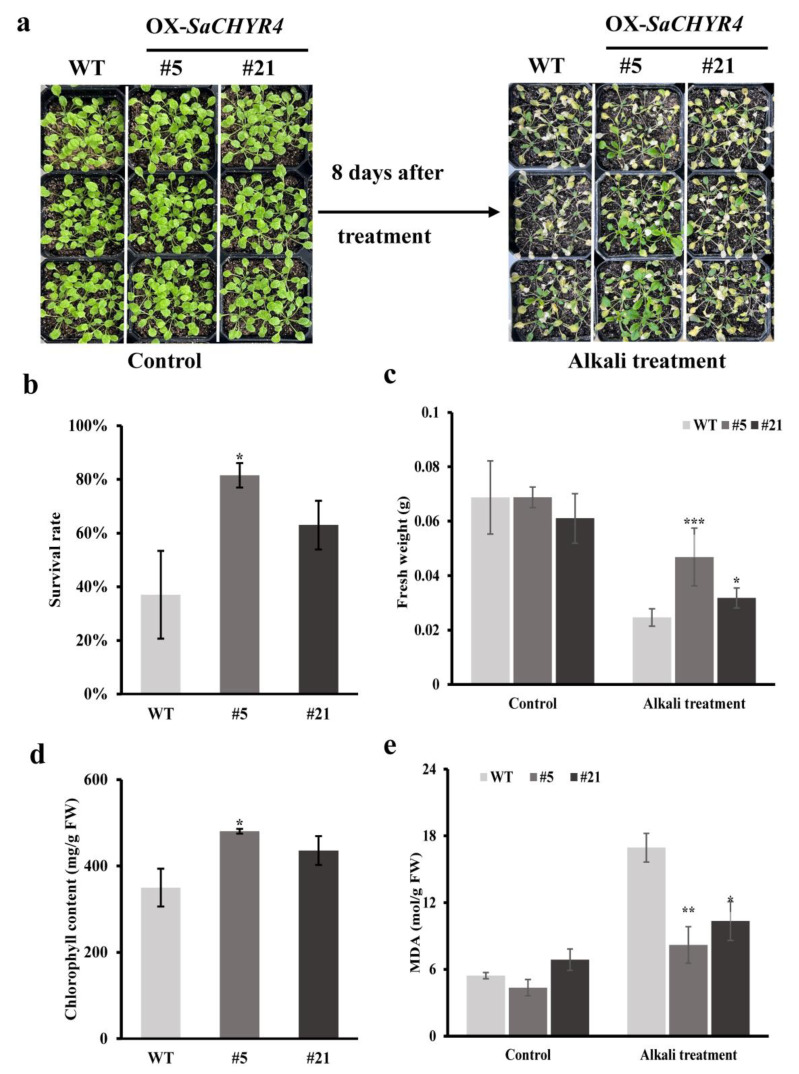
Effect of *SaCHYR4* overexpression on the seeding stage of *A. thaliana* under alkali stress. (**a**) Phenotypic state. (**b**) Survival statistics. (**c**) Fresh weight statistics. (**d**) Chlorophyll content. (**e**) Malondialdehyde content. Data represent means ± SD, with three biological replicates. GraphPad was used to calculate significance using one-way ANOVA. *, *p* < 0.05; **, *p* < 0.01, ***, *p* < 0.001.

**Figure 9 ijms-25-06173-f009:**
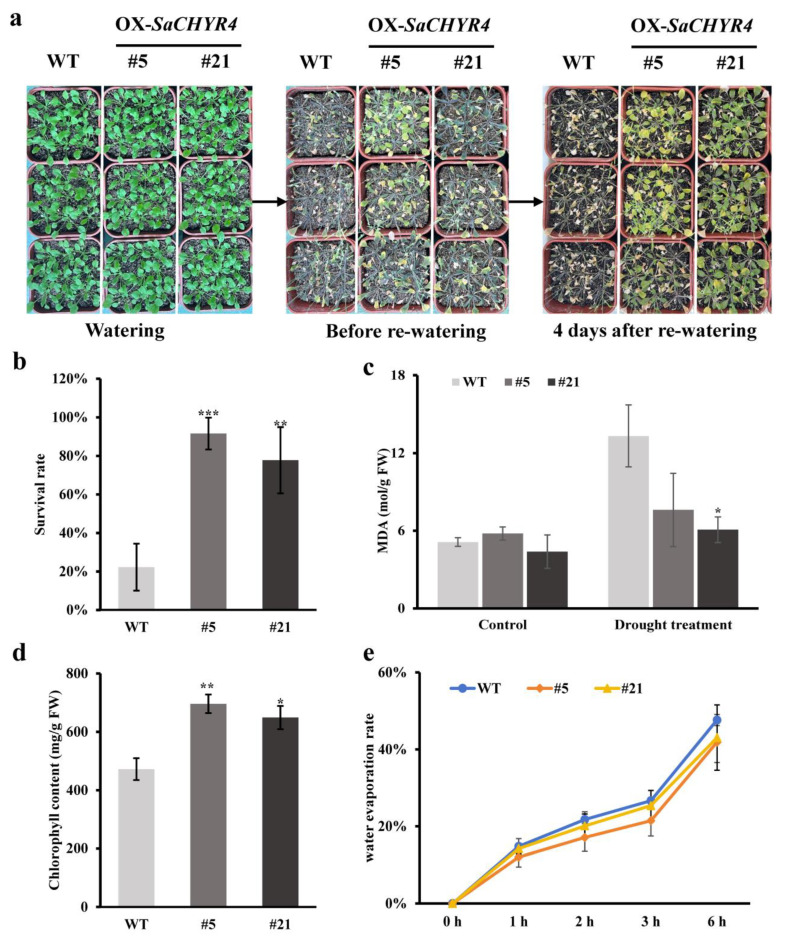
Effect of *SaCHYR4* overexpression on the seeding stage of *Arabidopsis thaliana* under drought stress. (**a**) Phenotypic state. (**b**) Survival statistics. (**c**) Malondialdehyde (MDA) content. (**d**) Chlorophyll content. (**e**) Transcriptional water loss. Data represent means ± SD, with three biological replicates. GraphPad was used to calculate significance using one-way ANOVA. *, *p* < 0.05; **, *p* < 0.01.

**Figure 10 ijms-25-06173-f010:**
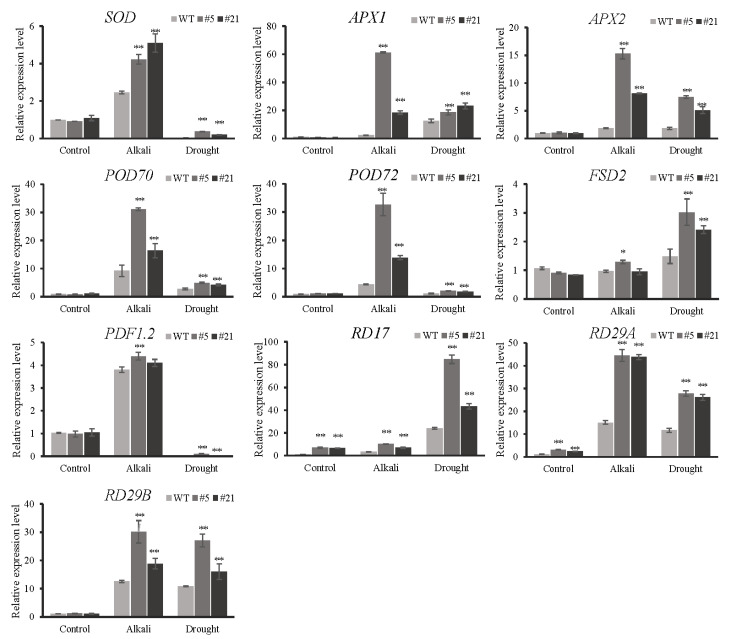
Relative expression levels of stress response genes in *Arabidopsis thaliana* under alkali and drought stress conditions. The relative expression level was calculated using the 2^−ΔΔCT^ method. Data represent means ± SD, with three biological replicates. GraphPad was used to calculate significance using one-way ANOVA. *, *p* < 0.05; **, *p* < 0.01.

**Table 1 ijms-25-06173-t001:** *CHYR* family genes identified in *Sophora alopecuroides* (*n* = 18).

No.	Gene	Gene ID	Subgroup	CDS Length (bp)	Protein Size (aa)	Molecular Weight (kDa)	pI	Predicted Localization
1	*SaCHYR1*	Sa01G051120	Clade Ⅰ	1020	339	39.06	7.83	Nucleus
2	*SaCHYR2*	Sa02G109750	Clade Ⅰ	813	270	31.14	6.94	Chloroplast
3	*SaCHYR3*	Sa05G228980	Clade Ⅰ	930	309	35.76	6.58	Chloroplast
4	*SaCHYR4*	Sa06G275940	Clade Ⅰ	930	309	35.69	6.41	Chloroplast
5	*SaCHYR5*	Sa07G314120	Clade III	3744	1247	140.60	5.52	Nucleus
6	*SaCHYR6*	Sa08G372170	Clade III	3744	1247	140.84	5.63	Nucleus
7	*SaCHYR7*	Sa09G431320	Clade II	816	271	31.15	6.10	Nucleus
8	*SaCHYR8*	Sa09G459760	Clade II	804	267	30.99	6.85	Chloroplast
9	*SaCHYR9*	Sa10G477850	Clade II	780	259	29.58	5.85	Nucleus
10	*SaCHYR10*	Sa10G510660	Clade II	804	267	31.13	6.87	Chloroplast
11	*SaCHYR11*	Sa11G521360	Clade III	3717	1238	139.64	5.67	Nucleus
12	*SaCHYR12*	Sa11G534300	Clade Ⅰ	927	308	35.46	6.89	Chloroplast
13	*SaCHYR13*	Sa12G563780	Clade III	3717	1238	139.69	5.67	Nucleus
14	*SaCHYR14*	Sa12G576360	Clade Ⅰ	927	308	35.42	6.81	Extracellular
15	*SaCHYR15*	Sa13G635820	Clade III	3864	1287	148.22	6.02	Chloroplast
16	*SaCHYR16*	Sa13G645440	Clade II	813	270	31.60	6.71	Chloroplast
17	*SaCHYR17*	Sa14G680390	Clade III	3864	1287	148.56	6.01	Chloroplast
18	*SaCHYR18*	Sa14G691080	Clade II	864	287	33.46	6.90	Chloroplast

## Data Availability

Raw data were deposited in the NCBI database under SRA accession numbers PRJNA636118 and PRJNA1077624.

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
