# Peer review of "Genome-Wide Identification of CHYR Gene Family in Sophora alopecuroides and Functional Analysis of SaCHYR4 in Response to Abiotic Stress"

_ijms, 2024, doi:10.3390/ijms25116173_

Round 1

Reviewer 1 Report

Comments and Suggestions for Authors

In the manuscript named “Genome-wide identification of CHYR gene family in Sophora alopecuroides and functional analysis of SaCHYR4 in response to abiotic stress”, Youcheng Zhu et al have performed genome-wide analysis of CHYR genes in Sophora alopecuroides, including bioinformatics analysis, RNA-seq analysis, RT-qPCR analysis, and transgenic analysis, they have found these genes with critical roles in response to abiotic stress. The findings were interesting to reads, and their conclusions were useful for molecular research about plant response to abiotic stress, the manuscript was also well prepared. However, there are some comments about it.

Major,

(1) The manuscript is missing all figures, it may be technological problem, please check them and update manuscript.

(2) There were some confusions in manuscript, in method section and results section. The details were listed as below.

Minor,

(3) Line 137, authors have described “Illumina sequencing was then performed using the PacBio sequencing technology”, I’m confused about Illumina sequencing using PacBio technology, please check it.

(4) “Clean reads were concatenated using Trinity and annotated using seven major databases (NR, NT, KO, SwissProt, PFAM, GO, and KOG)”, which meant authors had assembled these RNA-seq reads and annotated these unique transcripts, but this manuscript had performed genome-wide analysis using ref transcripts, I’m also confused on this description. In addition, authors have described “mapping reads with HISAT2”, please check it.

(5) evolview software should be added with refs.

(6) RT-qPCR (line 205) or qRT-PCR (line 214), please check it.

(7) “****was repeated thrice”, the “thrice” should be checked.

(8) “The selected Arabidopsis stress response genes included SOD, APX1***”, line 211, these words should be removed to over-expression analysis section, not here.

(9) “The germination stage was subjected to salt (1/2 MS medium containing 100 mM NaCl), alkaline (1/2 MS medium containing 7, 8, or 9 mM NaHCO3), and drought stress (1/2 MS medium containing 100 or 200 mM mannitol)”, the Arabidopsis have different mediums, while the abiotic stresses were also different, why? In addition, the drought had adopted mannitol, not PEG, why?

Author Response

Dear Reviewer:

       We are very grateful for your professional review of our article. Your concerns, raised several important issues that need to be addressed. Accordingly, we have made extensive corrections to our previous submission; the detailed corrections are listed below.

  1. The manuscript is missing all figures, it may be technological problem, please check them and update manuscript.

Response:

We rechecked and updated all the figures.

  1. There were some confusions in manuscript, in method section and results section. The details were listed as below.

Response:

Thanks for your valuable comments. We have added and updated the Methods and Results sections as per your comments.

  1. 3. Line 137, authors have described “Illumina sequencing was then performed using the PacBio sequencing technology”, I’m confused about Illumina sequencing using PacBio technology, please check it.

Response:

The original text in line 137 “Illumina sequencing was then performed using the PacBio sequencing technology” has been revised to “Sequencing was then performed using the PacBio sequencing technology (BGI Genomics Co., Ltd., Wuhan, China)”.

  1. “Clean reads were concatenated using Trinity and annotated using seven major databases (NR, NT, KO, SwissProt, PFAM, GO, and KOG)”, which meant authors had assembled these RNA-seq reads and annotated these unique transcripts, but this manuscript had performed genome-wide analysis using ref transcripts, I’m also confused on this description. In addition, authors have described “mapping reads with HISAT2”, please check it.

Response:

We annotated the obtained transcripts based on the genome of S. alopecuroides.

The original text starting in line 138 “Clean reads were concatenated using Trinity and annotated using seven major data-bases (NR, NT, KO, SwissProt, PFAM, GO, and KOG). Clean reads generated by an NGS platform were mapped to the genome sequence of S. alopecuroides (data not shown) using HISAT2 v2.2.1 with default parameters” has been revised to “Clean reads generated by an NGS platform were mapped to the genome sequence of S. alopecuroides (data not shown) using HISAT2 v2.2.1 with default parameters”.

  1. evolview software should be added with refs.

Response:

References regarding Evolview software have been added.

  1. RT-qPCR (line 205) or qRT-PCR (line 214), please check it.

Response:

In this study, we used reverse transcription quantitative PCR (RT-qPCR) to analyze gene expression levels. Thus, line 214 that read “qRT-PCR” has been revised to “RT-qPCR”.

  1. “****was repeated thrice”, the “thrice” should be checked.

Response:

Thank you for pointing this out. The original text in line 293 “the procedure was repeated thrice” has been revised to “each experiment was repeated three times”.

  1. “The selected Arabidopsis stress response genes included SOD, APX1***”, line 211, these words should be removed to over-expression analysis section, not here.

Response:

Thank you for pointing this out. We have completed the suggested revision. Specifically, the original text in line 211 and thereafter “The selected Arabidopsis stress response genes included SOD, APX1, APX2, POD70, POD72, FSD2, PDF1.2, RD17, RD29A, and RD29B. The NCBI web site (https://www.ncbi.nlm.nih.gov/tools/primer-blast/index.cgi?LINK_LOC = BlastHome) was used to design the RT-qPCR primers that were synthesized by Comate Bioscience Co., Ltd., Jilin Province” has been moved to the over-expression analysis section.

  1. “The germination stage was subjected to salt (1/2 MS medium containing 100 mM NaCl), alkaline (1/2 MS medium containing 7, 8, or 9 mM NaHCO3), and drought stress (1/2 MS medium containing 100 or 200 mM mannitol)”, the Arabidopsis have different mediums, while the abiotic stresses were also different, why? In addition, the drought had adopted mannitol, not PEG, why?

Response:

Thank you for your comments. In this study, to explore the tolerance of transgenic Arabidopsis thaliana to abiotic stress, we simulated salt, alkali, and drought stress at the seedling stage. Treatments utilizing multiple concentrations of these stressors were selected to ensure the reliability of the experimental results. Furthermore, mannitol was chosen instead of PEG to simulate drought stress because the medium with PEG was difficult to coagulate. Treatment concentrations and methods were based on previous studies.

Reviewer 2 Report

Comments and Suggestions for Authors

The article is of very good quality overall, so I can only congratulate the authors. This work demonstrates a very appropriate workflow and provides a lot of information about the CHYR gene family. However, I feel I must point out some aspects that I believe could help to round it off:

  1. The introduction contains a wealth of information and serves to contextualize the project very well, but perhaps it delves too deeply, especially from line 79 to line 105. Some of this information could be simplified or used in the discussion to provide a more in-depth explanation.

  2. Where has the S. alopecuroides genome database (unpublished) been obtained from? Have you sequenced it yourselves and are in the process of publishing it?

  3. Figures: The quality of figures 8, 9, and 10 needs to be improved as they appear blurry. Figure 4 would benefit from being enlarged. Please redo the figures as all other figures in the article are very good except for these. Correct this is a must.

  4. The phrase in line 365 "Methyltransferase and 'RING-type domain-containing proteins are reportedly involved in the regulation of plant responses to saline–alkali stress [33]'" is more suited for discussion rather than as a result. Please relocate this description to the discussion section.

  5. The discussion seems somewhat brief to me. I would add some of the points I've mentioned from the introduction and results, and expand the section on genes that may be regulated by these genes in relation to alkali and drought stress conditions, cause that is the main objetive of this work.

Author Response

Dear reviewer

We are very grateful for your professional review of our article. Your concerns raised several issues that need to be addressed. According to your valuable suggestions, we have made extensive corrections to our previous submission, and the detailed corrections are listed below.

  1. The introduction contains a wealth of information and serves to contextualize the project very well, but perhaps it delves too deeply, especially from line 79 to line 105. Some of this information could be simplified or used in the discussion to provide a more in-depth explanation.

Response:

Thank you for this very insightful comment. We moved lines 79 through 105 of the Introduction to the Discussion section.

  1. Where has the S. alopecuroides genome database (unpublished) been obtained from? Have you sequenced it yourselves and are in the process of publishing it?

Response:

The genome data of bitter bean was completed and assembled by our research group, but it has not been published.

  1. Figures: The quality of figures 8, 9, and 10 needs to be improved as they appear blurry. Figure 4 would benefit from being enlarged. Please redo the figures as all other figures in the article are very good except for these. Correct this is a must.

Response:

Figures 4, 8, 9, and 10 have all been reworked.

  1. The phrase in line 365 "Methyltransferase and 'RING-type domain-containing proteins are reportedly involved in the regulation of plant responses to saline–alkali stress [33]'" is more suited for discussion rather than as a result. Please relocate this description to the discussion section.

Response:

The text in line 365 “Methyltransferase and RING-type domain-containing proteins are reportedly involved in the regulation of plant responses to saline–alkali stress [33]” has been moved to the Discussion section.

  1. The discussion seems somewhat brief to me. I would add some of the points I've mentioned from the introduction and results, and expand the section on genes that may be regulated by these genes in relation to alkali and drought stress conditions, cause that is the main objective of this work.

Response:

We are grateful for your constructive suggestion. The following information has been added to the Discussion section.

Line 623-628: OsRZF34, a rice gene homologous to Arabidopsis AtCHYR1, increases stomatal opening and transpiration rate when overexpressed in Arabidopsis and rice, thereby enabling faster cooling of leaves. Populus euphratica overexpressing PeCHYR1 showed re-duced stomatal opening due to increased hydrogen peroxide (H2O2) content. Transgenic poplars showed high sensitivity to exogenous ABA and drought tolerance.

Line 634-654: Ring-type E3 ubiquitin ligases regulate not only enzyme activity via ubiquitination but also the expression of transcription factors (for example, WRKY and MYB) to regu-late stress response in plants. CHYR1 controls the binding of WRKY70 mediated by the phosphorylation of various cis-acting elements of SARD1, thereby regulating homeo-stasis between immunity and growth [39]; this implies that the E3 ubiquitin ligase functions by regulating the stability of its target proteins, and its function is regulated by protein phosphorylation and dephosphorylation. The RING-HC E3 ubiquitin-protein ligase SR1 regulates the expression of RAP2.2, a member of ERF-VII, by regulating the stability of the transcription factor WRKY33, thereby affecting plant hypoxia signal transduction in response to plant tolerance to flooding stress [40]. The Arabidopsis ubiquitin E3 ligases MIEL1 and MYB30 co-regulate the key ABA signaling pathway transcription factor ABI5, which is involved in seed germination [41]. The Ring-type E3 ligase MdMIEL1 in apples can mediate the ubiquitination and degradation of MdBBX7 through the 26S proteasome pathway, negatively regulating the response to drought stress [42]. MdSINA3 inhibits the promoting effects of MdBBX37 on leaf senescence by regulating MdBBX37 degradation. MdBBX37 interacts with MdABI5 in ABA signaling and MdEIL1 in ethylene signaling, affecting ABA- and ethene-mediated leaf senescence, respectively [43]. In this study, we predicted 31 proteins interacting with SaCHYR protein, including methyltransferase, ubiquitin-like domain-containing protein, and cytochrome P450. Methyltransferase and RING-type domain-containing proteins are reportedly involved in the regulation of plant responses to saline–alkali stress [29].

Line 663-666: Overexpression of SaCHYR4 in A. thaliana significantly improved alkaline and drought tolerance in seedlings. The expression levels of SOD, APX1, APX2, POD70, POD72, PDF1.2, RD17, RD29A, and RD29B in SaCHYR4 overexpression lines were significantly changed under alkali and drought stress. These findings provide a foundation for studying the role of the SaCHYR family genes in crop stress resistance.

Round 2

Reviewer 1 Report

Comments and Suggestions for Authors

Thanks for authors works, all comments have been well addressed, and the manuscript was well revised. There were two new minor comments, 1) the check marks in evolution figure could be removed, see figure 1. 2) the predication scores of protein structures could be removed in figure 5, which have badly displayed in figure 5. These scores could be listed in supplements.

Author Response

Dear Reviewer,

    We appreciate your latest comments on our article. We modified the previously submitted materials submitted. Please find the detailed description of our corrections below.

Thanks for authors works, all comments have been well addressed, and the manuscript was well revised. There were two new minor comments,

1)the check marks in evolution figure could be removed, see figure 1.

Response:

    According to your suggestion, we removed the evolution mark from Figure 1.

2) the predication scores of protein structures could be removed in figure 5, which have badly displayed in figure 5. These scores could be listed in supplements.

Response:

    Thank you for this remark. According to your suggestion, we removed the protein structure predication scores from Figure 5 and provided them as attachments (Table S3).
